# A Spectral Perspective on Deep Supervised Community Detection

## Abstract

In this work, we study the behavior of standard models for community detection under spectral manipulations. Through various ablation experiments, we evaluate the impact of bandpass filtering on the numerical performances of a GCN: we empirically show that most of the necessary and used information for nodes classification is contained in the low-frequency domain, and thus contrary to Euclidean graph (e.g., images), high-frequencies are less crucial to community detection. In particular, it is possible to obtain accuracies at a state-of-the-art level with simple classifiers that rely only on a few low frequencies: this is surprising because contrary to GCNs, no cascade of filtering along the graph structure is involved and it indicates that the important spectral components for the supervised community detection task are essentially in the low-frequency domain.

## 1 Introduction

Graph Convolutional Networks (GCNs) are the state of the art in community detection (Kipf & Welling, 2016). They correspond to Graph Neural Networks (GNNs) that propagate graph features through a cascade of linear operator and non-linearities, while exploiting the graph structure through a linear smoothing operator. However, the principles that allow GCNs to obtain good performances remain unclear. It is suggested in Li et al. (2018) that GCNs are eager to over-smooth their representation, which indicates they average too much neighborhood nodes and dilute classification information. The smoothing is generally interpreted as a low-pass filtering through the graph Laplacian, and finding a way to exploit high-frequencies of the graph Laplacian is an active research question (Oono & Suzuki, 2019). In contrast to this, our work actually suggests that, in the setting of community detection, graph Laplacian high-frequencies have actually a minor impact on the classification performances of a standard GCN, as opposed to standard Convolutional Neural Networks for vision, which are built thanks to image processing considerations.

Graph Signal Processing (GSP) is a popular field whose objective is to manipulate signals spectrum whose topology is given by a graph. Typically, this graph has a non-Euclidean structure, however many central theoretical results (Hammond et al., 2011) are based on an analogy with Euclidean, regular grids. For instance, a spectral component or frequency has to be understood as an eigenvalue of the Laplacian, yet it thus suffers from intrinsic issues such as isotropy (Oyallon, 2020). The principles of GSP are very appealing because they allow to use the dense literature of harmonic analysis, on graphs. Thus, this literature is at the core of many intuitions and drives many key ingredients of a GCN design, which evokes standard tools of signal processing: convolutions, shift invariance, wavelets, Fourier (Bronstein et al., 2017), etc. Here, we certainly observe several limits of this analogy in the context of community detection: for instance, we observe that discarding high-frequencies has a minor impact on a GCN behavior, because the spectrum of the graphs of the datasets that are used is essentially located in the low-frequency domain. This type of ideas is for instance core in spectral clustering algorithms.

Spectral clustering is a rather different point of view from deep supervised GCNs which studies node labeling in unsupervised contexts: it generally relies on generative models based on the graph spectrum. The main principle is to consider the eigenvectors corresponding to the smallest non-zero eigenvalues, referred to as Fiedler vectors (Doshi & Eun, 2020): those directions allow to define clusters, depending on the sign of a feature. Several theoretical guarantees can be obtained in the context of Stochastic Block Model approximation (Rohe et al., 2011). Our paper proposes

to establish a clear link with this approach: we show that the informative graph features are located in a low-frequency band of the graph Laplacian and do not need extra graph processing tools to be efficiently used in a deep supervised classifier.

This paper shows via various ablation experiments that experiments on standard community detection datasets like Cora, Citeseer, Pubmed can be conducted using only few frequencies of their respective graph spectrum without observing any significant performances drop. Other contributions are as follows: **(a)** First we show that most of the necessary information exploited by a GCN for a community detection task can actually be isolated in the very first eigenvectors of a Laplacian. **(b)** We numerically show that the high-frequency eigenvalues are less informative for the supervised community detection task and that a trained GCN is more stable to them. **(c)** We observe that a simple MLP method fed with handcrafted features allows to successfully deal with transdusctive datasets like Cora, Citeseer or Pubmed: to our knowledge, this is the first competitive results obtained with a MLP on those datasets.

We now discuss the organization of the paper: first, we discuss the related work in Sec. 2. We explain our notations as well as our work hypotheses in Sec. 3. Then, we study low-rank approximations of the graph Laplacian in Sec. 4.1. Finally, the end of Sec. 4 proposes several experiments to study the impact of high-frequencies on GCNs. A basic code is provided in the supplementary materials, and our code will be released on an online public repository at the time of publication.

## 2 RELATED WORK

**GCNs and Spectral GCNs**   Introduced in Kipf & Welling (2016), GCNs allow to deal with large graph structure in semi-supervised classification contexts. This type of model works at the node level, meaning that it uses locally the adjacency matrix. This approach has inspired a wide range of models, such as linear GCN (Wu et al., 2019), Graph Attention Networks (Veličković et al., 2017), GraphSAGE (Hamilton et al., 2017), etc. In general, this line of work does not consider directly the graph Laplacian. Another line of work corresponds to spectral methods, that employ filters which are designed from the spectrum of a graph Laplacian (Bruna et al., 2013). In general, those works make use of polynomials in the Laplacian (Defferrard et al., 2016), which are very similar to an anisotropic diffusion (Klicpera et al., 2019). All those references share the idea to manipulate bandpass filters that discriminate the different ranges of frequencies.

**Over-smoothing in GCNs**   In the context of GCN, Li et al. (2018) is one of the first papers to notice that cascading low-pass filters can lead to a substantial information loss. The result of our work indicates that the important spectral components for detecting communities are already in the low-frequency domain and that this is not due to an architecture bias. Zhao & Akoglu (2019); Yang et al. (2020) proposes to introduce regularizations which address the loss of information issues. Cai & Wang (2020); Oono & Suzuki (2019) study the spectrum of a graph Laplacian under various transform, yet they consider the spectrum globally and in asymptotic settings, with a deep cascade of layers. Huang et al. (2020); Rong et al. (2019b) introduce data augmentations, whose aim is to alleviate over-smoothing in deep networks: we study GCNs without this ad-hoc procedure.

**Spectral clustering and low rank approximation**   As the literature about spectral clustering is large, we mainly focus on the subset that connects directly with GCN. Mehta et al. (2019) proposes to learn an unsupervised auto-encoder in the framework of a Stochastic Block Model. Oono & Suzuki (2019) introduces the Erdös – Renyi model in the GCN analysis, but only in an asymptotic setting. Loukas & Vandergheynst (2018) studies the graph topology preservation under the coarsening of the graph, which could be a potential direction for future works.

**Node embedding**   A MLP approach can be understood as an embedding at the node level. For instance, Aubry et al. (2011) applies a spectral embedding combined with a diffusion process for shape analysis, which allows point-wise comparisons. We should also point Deutsch & Soatto (2020) that uses a node embedding, based on the spectrum of a quite modified graph Laplacian, obtained from on a measure of node centrality.

**Graph Scattering Networks (GSN)**   This class model explicitly employs band-pass based on the spectrum of a graph Laplacian and it is thus necessary to review it. Gao et al. (2019); Gama et al.

(2018; 2019) are a class of neural networks built upon an analogy with a Scattering Transform (Mallat, 2012). They typically rely on a cascade of wavelets followed by an absolute value: the objective of each wavelet is to separate multi-scale information into dyadic bandpass filters. This method relies heavily on each eigenvector of the Laplacian and the dyadic space is typically constructed from a diffusion process at dyadic intervals. Interferometric Graph Transform on the other hand relies on the concept of demodulation, which is clear in the context of Lie groups but unclear for community detection tasks (Oyallon, 2020).

**GCN stability** Stability of GCNs has been theoretically studied in Gama et al. (2019), which shows that defining a generic notion of deformations is difficult. Surprisingly, it was noted in Oyallon (2020) that stability is not a key component to good performances. The stability of GCN has also been investigated in Verma & Zhang (2019) but only considers neural networks with a single layer and relies on the whole spectrum of the learned layer. Keriven et al. (2020) considers the stability of GCNs, and relies on an implicit Euclidean structure: it is unclear if this holds in our settings. Sun et al. (2020) is one of the first works to study adversarial examples linked to the node connectivity and introduces a loss to reduce their effects. Zhu et al. (2019) also addresses the stability issues by embedding GCNs in a continuous representation. None of these work directly related a trained GCN to spectral perturbations.

## 3 FRAMEWORK

### 3.1 METHOD

We first describe our baseline model. Our initial graph data are node features $X$ obtained from a graph with $N$ nodes and an adjacency matrix $A$ with diagonal degree matrix $D$. We consider GCNs $f(X, A)$ as introduced in (Kipf & Welling, 2016), which correspond to GNNs that propagate features graph input $H^{(0)} \triangleq X$ through a cascade of layers, via the iteration:

$$H^{(l+1)} \triangleq \sigma\left(\tilde{A} H^{(l)} W^{(l)}\right),\tag{1}$$

where $\tilde{A} = \frac{1}{2}(I + D^{-1/2}AD^{-1/2})$, $\sigma$ a point-wise non-linearity and $W^{(l)}$ a parametrized affine operator. Note that if $\tilde{A} = I_N$, then Eq. 1 is simply an MLP, which makes its implementation simple. The $\frac{1}{2}$ factor is a normalization factor to obtain $\|\tilde{A}\| = 1$. In the semi-supervised setting, a final layer $f(X, \tilde{A}) \triangleq H^{(L)}$ is fed to a supervised loss $\ell$ (here a softmax) and $\{W^{(0)}, ..., W^{L-1}\}$ are trained in an end-to-end manner to adjust the label of each node. We note that for undirected graph, $\tilde{A}$ is a positive definite matrix with positive weights, which is understood as an averaging operator Li et al. (2018), as, ignoring $\tilde{D}$, we see that for some node features $X$, we have:

$$[(I_N + A)X]_i = X_i + \sum_{j \to i} A_{i,j} X_j .\tag{2}$$

We remark that multiple choices of averaging operators are possible: as briefly discussed in Appendix A.3 other formulations did not change our numerical conclusions, thus we decided to keep the simplest to be handled mathematically. We are interested in analyzing the properties of spectral approximations of $\tilde{A}$. We consider the decreasing set of eigenvalues $\Lambda = \{\lambda_k\}_{k \geq 0}$ of $\tilde{A}$, such that $\lambda_k \geq \lambda_{k+1}$, and we denote by $u_k$ the $k$-th eigenvector corresponding to $\lambda_k \in \Lambda$. We remind that $\Lambda \subset [0, 1]$ and that $\lambda_0 = 1$ can be interpreted as the lowest frequency of the graph Laplacian. Since the adjacency matrix is normalized, one basis of $\lambda_0$'s eigenspace is constituted by the constant vectors of 1 supported on each connected component. We then write:

$$\tilde{A}_{[k_1,k_2]} \triangleq \sum_{k_1 \leq k \leq k_2} \lambda_k u_k u_k^T ,\tag{3}$$

such that $\tilde{A} = \tilde{A}_{[0,N]}$. We are interested to study the degradation accuracy if we replace $\tilde{A}$ with $\tilde{A}_{[0,k]}$ or $\tilde{A}_{[k,N]}$ for some $0 < k < N$. The next section explains that under standard but oversimplifying assumptions, an approximation of the type $\tilde{A}_{[0,k]}$ is relevant for community detection tasks.

### 3.2 UNDERSTANDING LOW RANK APPROXIMATIONS FOR GCNS

We now justify our approach under the standard setting of the Stochastic Block Model (Abbe, 2017), and we will simply remind the reader of several elementary results. This model corresponds to a generative model that describes the interaction between $r$ communities $\{C_1, ..., C_r\}$. Assuming that two nodes $i, j$ belong to the communities $C_{r_i}, C_{r_j}$, an edge is sampled with a probability $p_{r_i, r_j}$ (Rohe et al., 2011). For the sake of simplicity, let us assume that $r = 2$, that the probability of an edge between two nodes $i, j$ is $p$ if those nodes belong to the same community and $q < p$ otherwise, and that both communities correspond to $|C|$ nodes. In this case, the unnormalized expected adjacency matrix is given by:

$$\begin{bmatrix} p & \cdots & p & q & \cdots & q \\ \vdots & & \vdots & \vdots & & \vdots \\ p & \cdots & p & q & \cdots & q \\ q & \cdots & q & p & \cdots & p \\ \vdots & & \vdots & \vdots & & \vdots \\ q & \cdots & q & p & \cdots & p \end{bmatrix}, \tag{4}$$

where we grouped in matrix block the nodes from the same community. We note that the two dominant eigenvectors are given by:

$$u_1 = [\overbrace{1, ..., 1}^{|C| \text{ times}}, \underbrace{1, ..., 1}_{|C| \text{ times}}] \text{ and } u_2 = [\overbrace{1, ..., 1}^{|C| \text{ times}}, \underbrace{-1, ..., -1}_{|C| \text{ times}}]. \tag{5}$$

Observe that the second eigenvector $u_2$ captures all the information about the two communities, through the sign of its coefficients. Here, the spectral gap (the ratio between the two dominant eigenvalues) is given by $0 \leq \frac{p-q}{p+q} < 1$ and ideally this spectral gap should be as large as possible for identifying the two communities. If the number of nodes is large, concentration results (Wainwright, 2019) imply that the empirical adjacency matrix concentrates around its expectation, and that under this assumption, a low-rank approximation $\tilde{A}_{[0,2]}$ captures most of the available information about the two communities. We illustrate this idea on Fig. 1. While these assumptions might not hold in practice, it justifies why low-rank approximations of a Laplacian are relevant in the setting of community detection and it explains why high-frequencies might not be as important as low-frequencies for supervised community detection task. The next section validates empirically this approach in the context of GCNs and simpler architectures.

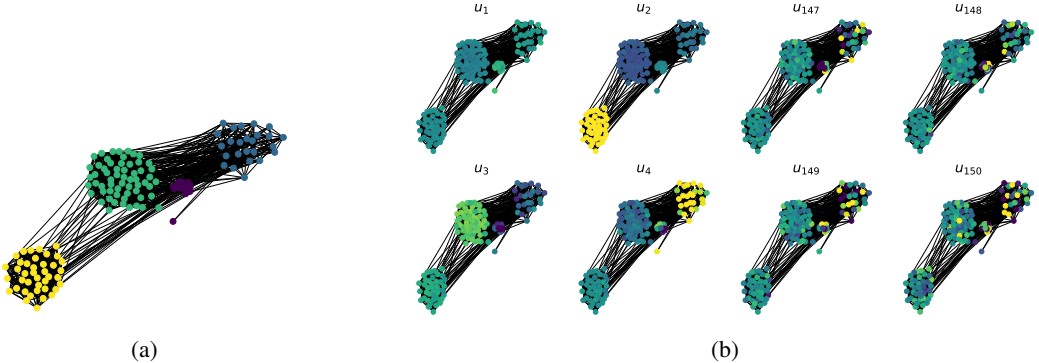

(a)                          (b)

Figure 1: Under a Stochastic Block Model with 4 communities (a), we represent the first Eigenvectors (b, left) and the last Eigenvectors (b, right). On the left figure, the colors stand for the communities. On the right, they stand for the values of the considered eigenvectors (the brighter the higher). A low rank approximation maintains the information related to the different communities.

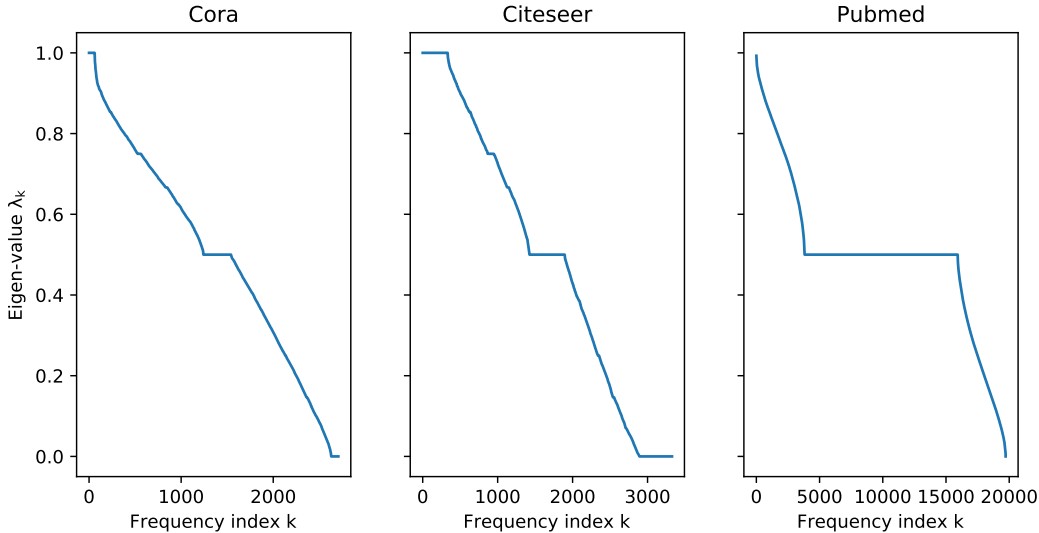

Figure 2: Spectrum $\Lambda$ of $\tilde{A}$. Note that the eigenvalues are decreasing, displayed with multiplicity. Observe that the decay of the spectrum is fast.

## 4 NUMERICAL EXPERIMENTS

Matching the previous work practice, we focus on the three classical benchmark dataset for community detection: Cora, Citeseer and Pubmed (Sen et al., 2008). The task consists in classifying the research topic of papers in three citation datasets. Those tasks are transductive, meaning all node features are accessible during training. We apply the full-supervised training fashion used in Huang et al. (2016), Chen et al. (2018), and Rong et al. (2019b) on all datasets in our experiments. Fig. 2 plots the 3 spectra $\Lambda$ in decreasing order for each of those datasets. The three datasets exhibit a significant spectral gap, which is aligned with the model of Sec. 1b. Note that Pubmed has one connected component, and that the decay of its spectrum is fast compared to Cora or Citeseer, which indicates a low-dimensional structure (Belkin & Niyogi, 2002). The statistics of each dataset are listed in the supplemental materials.

We choose $\sigma$ to be the ReLU non-linearity (Krizhevsky et al., 2012). Unless specified otherwise, the weights of our models (either GCN or MLP) are optimized via Adam, with an initial learning rate 0.01 and weight decay of 0.001, during 800 epochs. We use by default a dropout of 0.5. Our model consists in GCN layers with 2 hidden layers of size 128. In all experiments, we cross validate our hyper-parameters on a validation set, using an early stopping at epoch 400. Unless specified otherwise, each plot is obtained by an average over at least 3 different seeds.

### 4.1 LOW RANK APPROXIMATION

**GCN ablation**    Here, we consider the two projections $\tilde{A}_{[0,k]}$ and $\tilde{A}_{[N-k,N]}$, where $k$ is adjusted to retain only a portion of the spectrum. Via the GSP lens, those projections can be interpreted respectively as high-pass and low-pass filters. As explained in Sec. 1b, those projections will allow to study which frequency band is important for the community detection task. Fig. 3 and Fig. 4 report the respective numerical performance when considering the models $f(X, \tilde{A}_{[N-k,N]})$ and $f(X, \tilde{A}_{[0,k]})$ for some $k$.

On Fig. 3, we observe that retaining only very few frequencies (less than 10%) does not degrade much the accuracy of the original network. This does not contradict the observation of Li et al. (2018) which studies empirically the over-smoothing phenomenon, as our finding indicates that a GCN uses mainly the low frequency domain. For Pubmed, using almost all the high-frequencies is required to recover the original accuracy of our model. Interestingly, deeper GCNs seem to benefit from the high frequency ablation, yet their accuracy remains below their shallow counter-part and

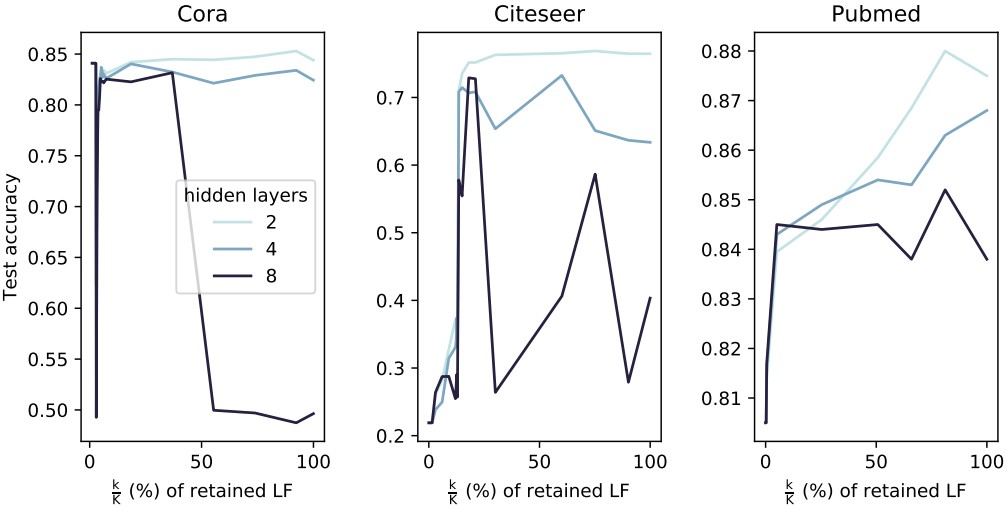

Figure 3: Test accuracies reached by a GCN as a function of the Low-Frequencies (LF) band $[\lambda_k, \lambda_0]$ retained by $\tilde{A}_{[0,k]}$, for various depths (100% corresponds to the full spectrum, including low frequencies). This figure indicates that informative component for the community detection task are located in the low frequency domain.

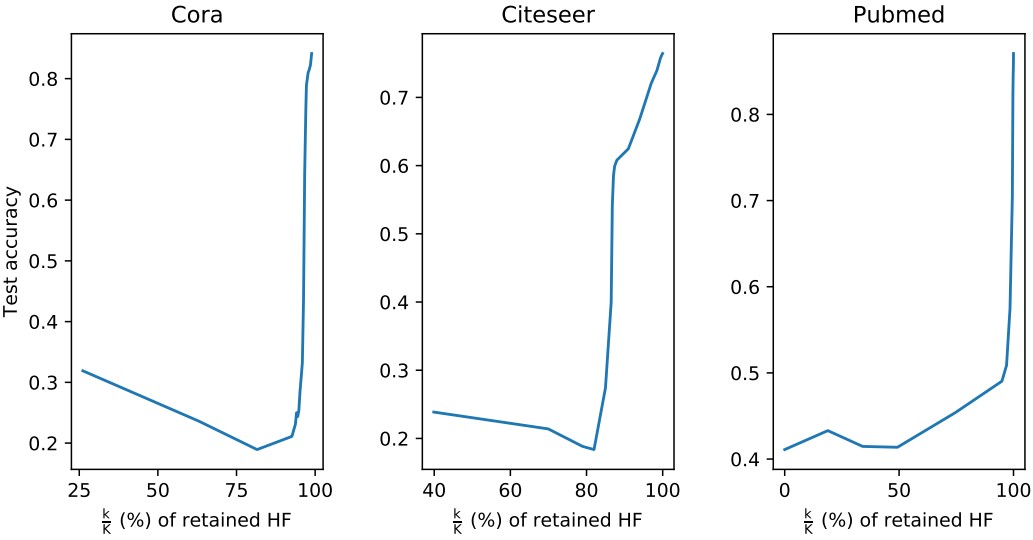

Figure 4: Test accuracies reached by a GCN as a function of the High-Frequencies (HF) band $[\lambda_N, \lambda_{N-k}]$ retained by $\tilde{A}_{[N-k,N]}$, for a GCN of depth 2 (100% corresponds to the full spectrum). This figure indicates that high-frequencies are less informative for a community detection task.

they are still difficult to train, as shown on Cora. This instability to spectral perturbations is further studied in Sec 4.2. Fig. 4 indicates that the major information for supervised community detection is contained in the low frequencies: dropping the latter leads to substantial accuracy drop, even for a shallow GCN.

**MLP ablation** We further study how informative low-frequencies are for community detection tasks via a spare ablation experiment based on a MLP. We augment each graph features $X$ through the concatenation $X_k = [X, u_1^T, ..., u_k^T]$ of the first $k$ eigenvectors. For a fair evaluation, we used exactly the same hyper-parameters as for the experiments above. Fig. 5 reports the accuracy of our MLP trained on $X_k$ as a function of $k$. We see that using a fraction $\frac{k}{K} \leq 20\%$ of the eigenvectors

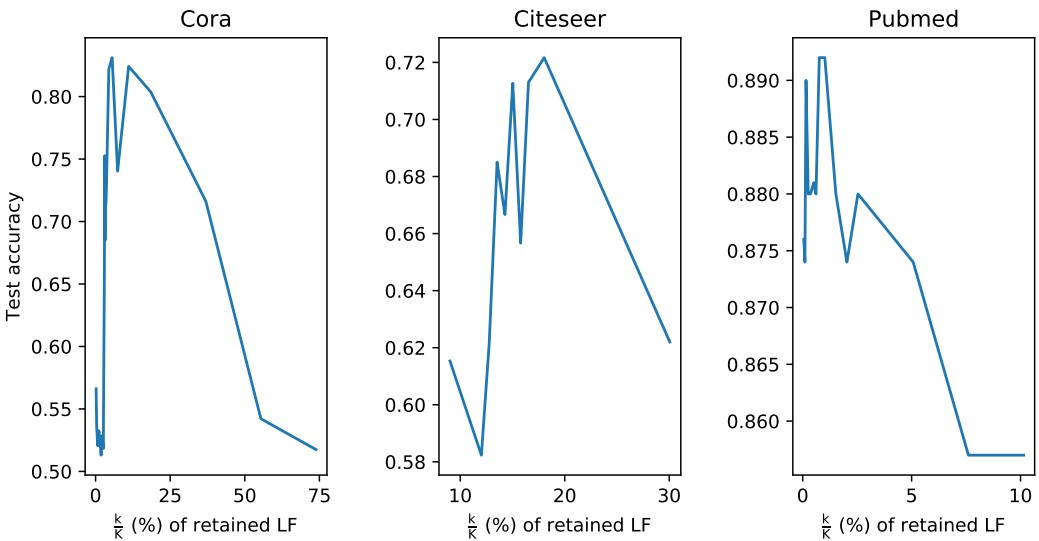

Figure 5: Accuracy of MLPs trained on $X_k$, according to the Low-Frequency band $[\lambda_k, \lambda_0]$ retained. We note that selecting a narrow low-frequency band can lead to competitive accuracies.

allows to recover the performance of a GCN trained end-to-end. More surprisingly, we note that as $k$ increases, the accuracy drops, which indicates that high-frequencies behave like a residual noise that is not well filtered by a MLP, and rather overfitted. This is particularly true for Pubmed, which has a fast spectral decay and for which the low frequencies seem to be the most informative. This experiment emphasizes that a low rank approximation of the Laplacian of the graph of a community detection task can be beneficial to a MLP classifier.

Table 1: Comparison of various models on Cora, Citeseer and Pubmed.

| Method | Data augmentation | Cora | Citeseer | Pubmed |
|---|---|---|---|---|
| GCN (Rong et al., 2019b) | No | **86.6** | **79.3** | 90.2 |
| Fastgcn (Chen et al., 2018) | No | 86.5 | - | 88.8 |
| MLP on $X$ | No | 74.0 | 73.3 | 89.1 |
| MLP on $\tilde{X}_k$ (ours) | No | **86.6** | 77.3 | **91.4** |
| DropEdge (Rong et al., 2019b) | Yes | 88.2 | 80.5 | 91.7 |
| (Huang et al., 2018) | Yes | 87.4 | 79.7 | 90.6 |

**Boosting MLP performances** We perform a hyper-parameter grid search on each dataset to investigate MLP strengths further, and report the case giving the best accuracy on each validation set. We summarize our findings in Tab. 1. In particular, one uses a fraction 5.9%, 15.0% and 0.7% of the spectrum respectively on Cora, Citeseer and Pubmed in order to obtain our best performances. We note that a MLP trained solely on $X$ already outperforms the approach of Chen et al. (2018), without relying on a graph structure. More details on the methodology are provided in the supplementary material. This simple model is highly competitive with concurrent works like Rong et al. (2019b); Huang et al. (2018); Chen et al. (2018), and in particular with vanilla GCNs. Note also that our method does not incorporate any data augmentation procedure such as Rong et al. (2019a), and thus a performance gap still remains. We could also potentially incorporated data augmentation procedures, at the price of an extended computation time. We conclude that GCNs do not compute more complex invariants than a MLP fed with low-frequencies, in the context of community detection.

**Note on the computational overhead** The MLPs introduced above are of interest if the corresponding graph topology is fixed, with a large graph, and high connectivity. Indeed, using a MLP

allows to easily employ mini-batch strategies and the training data can be reduced according to the fraction of low frequencies being kept: an exact $k$-truncated SVD has a complexity about $\mathcal{O}(kN^2)$. We note that fast $k$-truncated $\epsilon$-approximate SVD algrotithms for sparse matrix exist (Allen-Zhu & Li, 2016): if $\rho$ is the number of non-zero coefficients of $\tilde{A}$, the complexity can be about $\mathcal{O}(\frac{k\rho}{\epsilon} + \frac{k^2 N}{\epsilon})$.

## 4.2 STABILITY TO HIGH FREQUENCIES

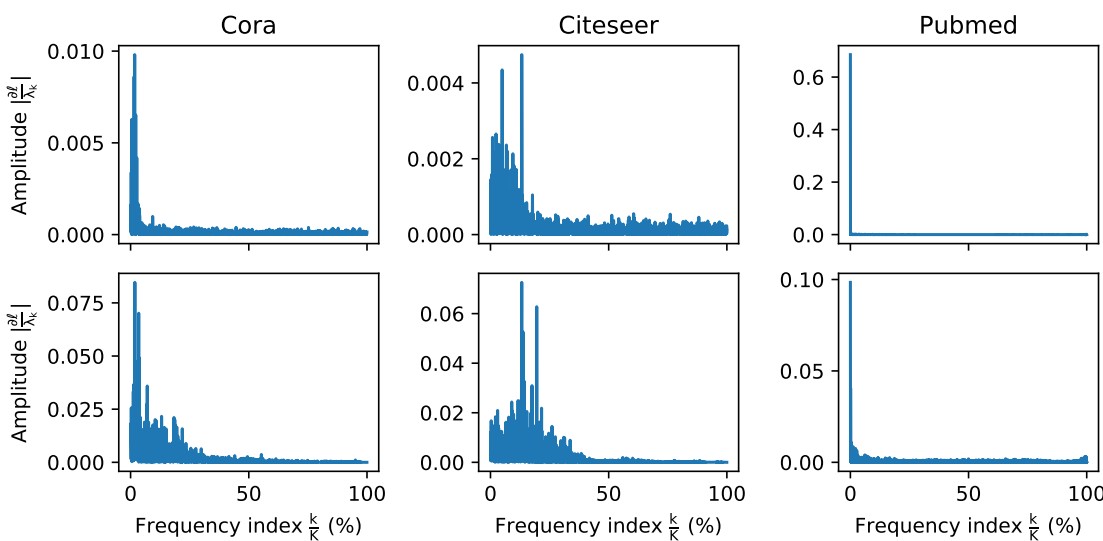

Figure 6: The (top) and (bottom) figures corresponds to $|\frac{\partial \ell}{\lambda_k}|$ of the same model taken respectively at the initialization and the end of training. Note that high frequencies are more stable than low frequencies for the three datasets.

We now study the stability of a GCN w.r.t. spectrum perturbations. In the case of image processing, it is standard that low-frequencies almost do not affect the classification performances and that perturbations of high-frequencies lead to instabilities. We would like to validate that this principle does not hold here, and to do so, we consider $\nabla_\Lambda \ell$, which is the gradient w.r.t. every singular value $\lambda_k$. Small amplitudes of $|\frac{\partial \ell}{\partial \lambda_k}|$ indicate more stable coefficients. Fig. 6 plots the amplitude of the gradient w.r.t. $\lambda_k$ at the initialization of a GCN and after training. First, we note that a GCN is more sensitive to spectral perturbations after training, which is logical because the GCN adapts its weights to the specific structure of a given graph. After training, we remark that the high-frequency perturbations have a small impact compared to the low-frequency perturbations on the three datasets, except for Pubmed which has dominant low-frequencies. This is consistent with our previous findings.

## 4.3 SMOOTHNESS AND SELF-LOOPS

As proposed by Kipf & Welling (2016), we now study how the self-loop of $\tilde{A}$ affects our GCN training procedure and in particular, we study it through the lens of spectral analysis. We consider:

$$\tilde{A}_\eta \triangleq \frac{1}{1+\eta}\left(\eta I_N + \tilde{A}\right),\tag{6}$$

where $\eta \in [-\frac{1}{2}, +\infty[$ can be understood as a smoothness parameter. Note that for $\eta < 0$, the spectrum of $\tilde{A}_\eta$ is not necessarily positive and we renormalize $\tilde{A}_\eta$ such that $\|\tilde{A}_\eta\| = 1, \forall \eta$. Here, $\tilde{A}_\eta$ enhances the high-frequencies of the averaging while allowing to balance a trade-off between a smoothing and an identity operator, which can be here observed as

$$\tilde{A}_\eta \underset{\eta \to \infty}{\sim} I \qquad \text{and} \qquad \tilde{A}_0 = \tilde{A}.\tag{7}$$

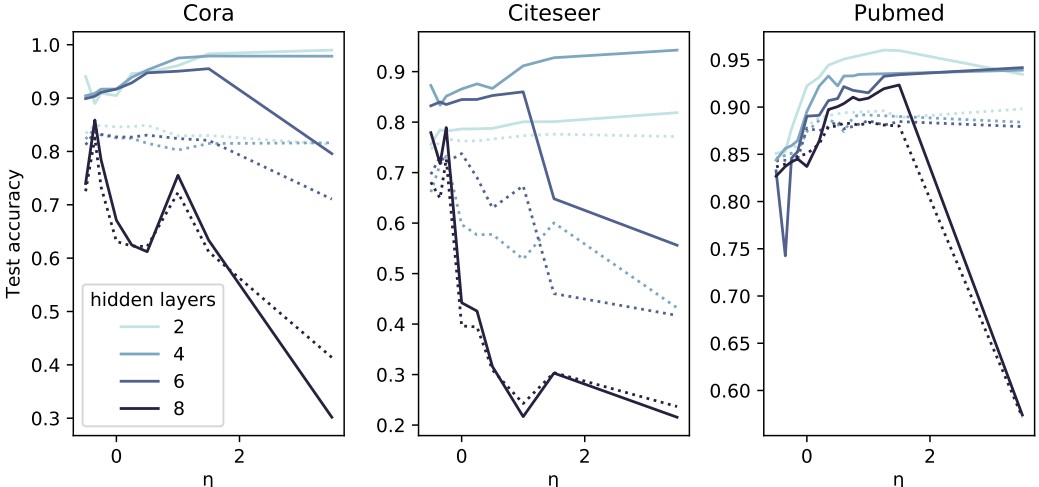

Figure 7: Test (dashed line) and train (plain line) accuracy according to $\eta$ for various depth of a GCN. We observed marginal improvements when varying $\eta$.

As observed in Li et al. (2018), oversmoothing phenomena occur because one propagates an input through a cascade of $n$ smoothing operators. Here, we note that a Taylor expansion reveals asymptotically this trade-off for large $\eta$:

$$\tilde{A}_\eta^n = \sum_{\lambda_k=1} u_k u_k^T + \sum_{\lambda_k<1} u_k u_k^T - \frac{n(1-\lambda_k)}{\eta} u_k u_k^T + \mathcal{O}(\frac{1}{\eta^2}). \tag{8}$$

Thus depending on the nature of $u_k$, the degree of averaging can be adjusted. The details of the calculations are given in App. A.4. Fig. 7 indicates that it is possible to adjust the degree of smoothing to train deeper networks (in the degenerate setting $\eta < 0$) and to slightly boost the accuracy of a GCN. We report both the training and testing accuracies to emphasize the impact of $\eta$ on the training dynamics of a GCN. Citeseer exhibits a significant generalization gap for large values of $\eta$ and a depth of 4: high-frequencies deteriorate the generalization properties. We note that varying $\eta$ allows to improve the training accuracy of deeper GCNs, though larger $\eta$ emphasizes higher-frequencies which leads to a significant drop in accuracy. We finally observe that selecting $\eta$ via the best accuracy on a validation set allows minor boosts in performances: of the order of 1.4% ($\eta = 1$) and 1.5% ($\eta = 0.5$) for a GCN of depth 2 respectively on Citeseer and Pubmed.

## 5 CONCLUSION

In this work, we have studied the classification performance of a GCN if applying low-pass and high-pass filters, in the context of community detection. Our finding is that by design, a GCN mainly relies on the low-frequencies, and that the high-frequencies have less impact on this task. Then, we are able to design MLPs that rely simply on a few eigenvectors of the graph Laplacian that are competitive with deep supervised graph approaches. We also study the stability of a GCN w.r.t. spectral perturbations, and show that they are more robust to high-frequency, which is counter-intuitive when compared to vanilla CNNs.

Our work indicates that not only more difficult graph benchmarks are necessary, but also benchmarks whose optimal model would rely on the high-frequencies of a graph Laplacian. It also shows that standard GSP tools such as graph wavelets might be simplified to low-pass filters, in the context of community detection, as we observed that the high-frequency does not bring a significant amount of information, and can even be interpreted as a residual noise, in this particular setting. Note also that our methodology can help to identify if an accuracy improvement of a given algorithm is due to a better processing of high frequencies.

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

# A APPENDIX

## A.1 DATASET STATISTICS

Table 2: Dataset Statistics

| Datasets | Nodes | Edges | Classes | Features | Traing/Validation/Testing split |
|---|---|---|---|---|---|
| Cora | 2,708 | 5,429 | 7 | 1,433 | 1,208/500/1,000 |
| Citeseer | 3,327 | 4,732 | 6 | 3,703 | 1,812/500/1,000 |
| Pubmed | 19,717 | 44,338 | 3 | 500 | 18,217/500/1,000 |

## A.2 HYPER-PARAMETER DESCRIPTION

Table 3: Hyper-parameter Description

| Hyper-parameter | Description |
|---|---|
| lr | learning rate |
| hidden layers | the number of hidden layers |
| weight-decay | L2 regulation weight |
| dropout | dropout rate |
| frequencies | the number of low frequencies to add |
| eigenvector features normalization | whether to normalize the new MLP features in line (per nodes) or column (per features) |

## A.3 ON THE CHOICE OF THE NORMALIZATION

GCNs defined in Kipf & Welling (2016) don't exactly used the first order Laplacian approximation, but introduce a normalization trick : $\tilde{A} = (D + I_N)^{-\frac{1}{2}}(A + I_N)(D + I_N)^{-\frac{1}{2}}$. Using the standard hyper-parameters defined in Section 4, we averaged our results on 3 seeds and observed no significant difference across datasets, as shown in Fig. 8.

## A.4 TAYLOR EXPANSION

We recall that $\tilde{A}_\eta = \frac{1}{1+\eta}(\eta I_N + \tilde{A})$. Thus an eigenvector $u_k$ of $\tilde{A}$ corresponding to the eigenvalue $\lambda_k$ is an eigenvector of $\tilde{A}_\eta$ corresponding to the eigenvalue $\frac{\eta+\lambda_k}{1+\eta}$. Therefore, we have:

$$\tilde{A}_\eta = \sum_{\lambda_k} \frac{\eta + \lambda_k}{1 + \eta} u_k u_k^T ,$$

And thus,

$$\tilde{A}_\eta^n = \sum_{\lambda_k} \left(\frac{\eta + \lambda_k}{1 + \eta}\right)^n u_k u_k^T$$

$$\tilde{A}_\eta^n = \sum_{\lambda_k=1} u_k u_k^T + \sum_{\lambda_k<1} \left(\frac{\eta + \lambda_k}{1 + \eta}\right)^n u_k u_k^T$$

$$\tilde{A}_\eta^n = \sum_{\lambda_k=1} u_k u_k^T + \sum_{\lambda_k<1} u_k u_k^T - \frac{n(1 - \lambda_k)}{\eta} u_k u_k^T + \mathcal{O}(\frac{1}{\eta^2})$$

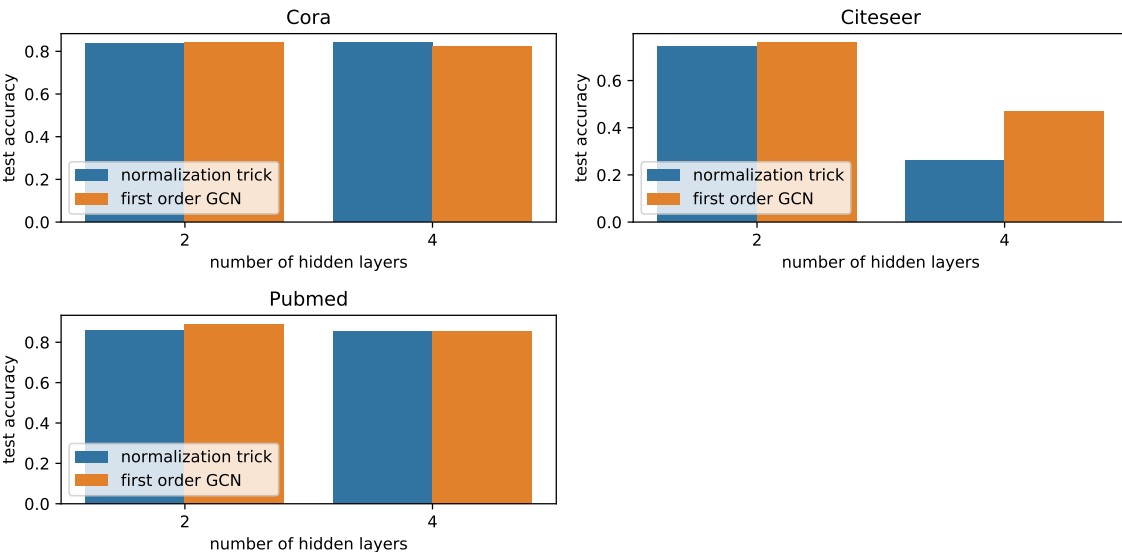

Figure 8: Comparison between the standard Laplacian first order normalization and the GCN normalization trick

