# OpenReview forum: "A Spectral Perspective on Deep Supervised Community Detection"
_ICLR.cc/2021/Conference — Reject_

### Official Review · AnonReviewer1 · 2020-10-21
**A nice analysis of GCN models**

**Rating:** 6
**Confidence:** 2

**Review:**

Summary:
The work presents an interesting analysis of GCN models under spectral manipulations and relates the performance of GCNs through bandpass filtering. The authors demonstrate that GCNs mainly rely more on low-frequencies rather than high-frequencies which is contrary to what is observed in signal processing. For this, the authors use band-pass filters which allow only a portion of the spectrum to be utilized by the GCN model. The major findings are as follows:
1. The high-frequency eigenvalues are less informative and perturbing them does not have much effect on GCN’s output. This supports the over-smoothing phenomena empirically observed by Li et al. 2018
2. A simple MLP with few initial eigenvectors as additional features outperforms several existing GCN models. Moreover, including higher eigenvectors degrades performance as it adds noise which could not be well filtered by MLP.


Questions:
1. The reason why GCN model with 8 hidden layers gives poor performance is not properly justified. It would be great if authors could provide more intuition behind it. Can over-parameterization can be a factor for such poor generalization?
2. Can the authors provide some explanation behind the sudden jerks observed in Figures 2 and 4?

Typos:
1. In the related work section (para 2 and 3) change ‘overs-moothing’ to ‘over-smoothing’ and ‘litterature’ to ‘literature’.

---

> ### Author Response · Authors · 2020-11-13
> **Re: A nice analysis of GCN models**
>
> Dear Reviewer,
>
> We thank you for your positive review and your time. We have emphasized in this new version of the paper that we have a specific focus on community detection tasks, and we added Sec 3.2 which justifies the well-foundedness of our method thanks to Stochastic Block Models. Those elements are aligned with the summary of the reviewer. We address your questions below.
>
> 1.  This is an interesting question raised also by AnonReviewer3. Deeper GCNs are known to be more difficult to train (Li et al 2019), as their training is known to be more unstable than their shallow counterpart. We could have partially solved this issue by performing more parameter tuning for deeper GCNs, but we believe it would make the comparisons between different models and datasets less clear. Note that works like (Rong et al 2019) draw similar conclusions to ours.
>
> 2.  We believe Fig. 6 gives a partial answer to this question by measuring the variation in amplitude if one perturbs the spectrum of our graph: one sees that some frequencies might affect significantly and heterogeneously the loss of a given model on a given dataset. Those instabilities are purely dependent on the data which are used and how the spectrum structures the communities.
>
> Thank you very much for reporting those typos that we corrected.
> We would be extremely interested to have your view on our modifications and clarifications and we thank you again for your time.

---

### Official Review · AnonReviewer3 · 2020-10-27
**Novelty  is limited**

**Rating:** 4
**Confidence:** 4

**Review:**

Summary:
 The article analyzes GCNs from spectral viewpoint, and discusses the performance of GCNs with respect to spectral filtering. The paper shows by experimentation, that the performance of GCNs mainly depend on low frequencies (lower end of the spectrum/eigen-pairs). It then shows that an MLP with low frequency information (Eigen-pairs) performs very well in graph tasks. Aspects such as smoothness and high frequency ablations are also studied.

--------------------
Strengths:
1. Paper presents a study of GCNs from spectral perspective.
2. Paper makes few interesting observations about the influence of low and high frequency components of the Graph Laplacian on GCN performance.
3. MLP with spectral information is investigated for graph tasks.
--------------------
Weakness:
1. Novelty in the study is limited.
2. Experimental results are inadequate.
3. Presentation is poor.
--------------------
Details:
I have the following comments about the paper:

1.  Novelty is limited: The novelty of this study is unclear to me. The fact that the GCN model of (Kipf and Welling, 2016) is based on low pass filtering is well-known. Note that, their model is simply a degree-one approximation of the low pass Chebyshev filter approach of (Defferrard et al., 2016). The original objective of spectral GCNs have been low pass (localized) filtering as proposed by (Bruna et al., 2013). The study in (Li et al., 2018) also further establishes these spectral results. Hence, the novelty of the paper is limited.

2. The experiment results are inadequate. The paper considers just 3 datasets (all three have similar spectrum), and it is hard to make conclusions with the few results presented. The results in Figure 2, do not actually support the conclusions. The effects of retaining more LF components is not clear, especially for deeper networks.
Results on more datasets with different spectral distributions would make the study more conclusive.

3. The paper is poorly written, and needs to be checked for language consistency throughout. I recommend having the paper proof-read by a native speaker.
Notation is inconsistent. Both upper and lower case letters are used for matrices.

4. Cost: Note that GCN is a very simple model that is computationally inexpensive. It requires just few matvecs (# of features) at each layer to implement.
The proposed MLP requires eigenvectors, i..e, requires a complete Eigen-decomposition of the Laplacian, that has a cubic K^3 cost. Computational complexity of the compared methods should be discussed.

Minor Comment:
i. Page 2, "one of the first paper" --> one of the first papers
"litterature", "overs-moothing" and many more incorrect spellings throughout

---

> ### Author Response · Authors · 2020-11-13
> **Re: Novelty is limited**
>
> Dear Reviewer,
>
> We thank you very much for your detailed review and insightful comments. We would like to highlight that, thanks to the reviewer feedback, we have certainly narrowed the scope of this paper and we now focus the presentation mainly on community detection tasks, because we believe our findings mainly apply to this setting (see Sec3.2). In particular, contrary to concurrent works that study low-pass filtering phenomena due to architecture biases, our analysis is relevant to the data, and we show that the important information in the case of community graphs, even in supervised contexts, is located in the low-frequency domain. Claiming that a model operates on low frequencies is different from claiming that high-frequencies are over-smoothed (and thus discarded). We answer the points you raised below.
>
> 1. Novelty is limited. As pointed above, the papers (Li et al 2018, Rong et al 2019, Bruna ) try to build Graph Neural Networks that demodulate high-frequencies in the low frequency domain. Our contribution is to notice that, in fact, due to the structure of Cora, Citeseer and Pubmed (which are the standard datasets used in Li et al 2018, Rong et al 2019) the important information for detecting communities is located in the low-frequency domain (Sec 3.2-4.1-4.2). It implies that other datasets, for which high-frequencies are more crucial,might be more relevant to study the oversmoothing phenomenon. We thus believe that our work is an important contribution in this direction because we justify in which context Cora, Citeseer or Pubmed are appropriate benchmarks.
>
> 2. The experiment results are inadequate. We would like to re-emphasized that the paper is now mostly focused on community detections tasks, and we believe the usefulness of a low-rank approximation is specific to this task, as explained in the new Sec3.2. As Cora, Citeseer and Pubmed are the most standard benchmarks used in GCN, we believe that they are the most relevant to work with.
> We agree that our spectral ablations show that deeper GCNs are more difficult to train, but the general tendency is that discarding high-frequencies helps them during training. Yet, this is more a side experiment and again, our objective is to give some insights on the nature of the data, rather than finding solutions to the over-smoothing phenomenon.
>
> 3. The paper is poorly written. We do apologize and we used the possibility provided by Openreview to rewrite the unclear parts of the paper and double-check them with a native speaker. We further used the notations introduced by (Kipf & Welling 2016). We hope that the paper now reaches a decent level of writing.
>
> 4. Cost. We added a paragraph about the cost. Note that the method is actually quadratic in the size of the graph because one does not need (as shown Fig 5) to perform a full exact SVD, however, as now pointed in the paper, approximate algorithms exist for sparse truncated SVD which are significantly faster(see: [https://arxiv.org/pdf/1607.03463.pdf](https://arxiv.org/pdf/1607.03463.pdf) ). To give a more precise order of magnitude, the calculation of 100 eigenvalues takes respectively 2.5s on Cora, 0.5s on Citeseer and 14s on Pubmed on a standard laptop.
>
> We would be extremely interested to have your view on our modifications and clarifications and we thank you again for your time.

---

### Official Review · AnonReviewer4 · 2020-10-28
**Official Blind Review #4**

**Rating:** 3
**Confidence:** 4

**Review:**

This paper aims to study how GCN will behave under spectral perturbations/manipulations. The empirical numerical analysis on three benchmark datasets (cora, citeseer, pubmed) show that most of the necessary information is contained in the low-frequency domain. Based on that, the author propose to expand the node feature matrix with the eigenvectors corresponding to low-frequency domain and apply MLP on this new feature matrix. Experimental results show that the proposed method outperforms vanilla GCN and achieve comparable results on pubmed with other baselines.

I think the paper has a pretty good start point: understanding how the spectrum of adjacency matrix will affect the behavior of GCN. But I think the manuscript is loosely written and I can hardly follow it. I do have a lot of confusion about this paper and hope that the authors can clarify them.

- In general, there are too many typos and grammar errors which make the manuscript hard to read.
- Sec 1: in the 1st paragraph, what does 'graph principles' mean? I cannot recall a clear definition of this term. I think the authors could clarify it before using it formally.
- Other contributions in Sec 1:
  * (a), I think the empirical results in the paper show that retaining a small portion of low-frequencies is enough for achieving good classification accuracy. From the results and your analysis, I can hardly find a clear conclusion that links to it (i.e., the very first eigenvector is most informative);
  * (b) and (e) are somehow connected since both are mentioning about GCN's behavior over manipulating high frequencies;
  * (c), I actually did not quite understand where the clear link is. I would suggest the authors clarify it more clearly in the manuscript;
  * Just a minor thing, what exactly does 'informative' mean, greatest change in loss function or greatest change in test accuracy? Better make it clear.
- Sec 3:
  * Notations are quite messy. A matrix can be denoted using italic lowercase letter, italic uppercase letter, calligraphic uppercase letter, bold italic uppercase letter. It is very confusing when reading the paper, especially when some scalars are also denoted using the same convention.
  * I did not find any framework in this section, which contradicts the last paragraph in Sec 1.
- Sec 4:
  * Pubmed should have only 1 connected components. In the following sentence, what does 'not necessarily fully supported on ...' mean?
  * In Sec 4.1, is there any intuition or theoretical justification for the projection operation used in band-pass filter?
  * In Figure 2, it seems that Cora has a very sharp performance drop (~0.82 -> ~0.5) when x-axis is nonzero, is there any insightful explanation on that?
  * Does the content of 'MLP ablation' correspond to your proposed method? This is the only place I can find clues about your method. I understand that the goal of this paper is show a MLP can perform better or comparable with GCN that perform message passing. But I believe you still need topology information to get those eigenvectors. Is there any insight or theoretical concern about simple concatenation instead of linear transformations like $e^T e x$?

---

> ### Author Response · Authors · 2020-11-13
> **Re: Official Blind Review #4**
>
> Dear Reviewer,
>
> We thank you very much for your detailed review and insightful comments. Following your suggestions, we have modified the presentation of the paper: (a) we have re-emphasized that we have a particular focus on community detection tasks. Now, we believe that the message we deliver is extremely clear: in a supervised context, for community detection tasks, the graph Laplacian has a low dimensional structure. (b) We have in particular added a Sec 3.2 that emphasizes why our results are specific to community detection, by re-explaining how stochastic block models can explain our results. (c) We clarified our contributions. We used fully the possibility given by Openreview to perform some minor modifications in the manuscript, which emphasize our initial starting point.
>
> Contributions: We simplified and clarified our contributions. The main message of our work is that experiments on standard community detection datasets like Cora, Citeseer, Pubmed can be conducted using only a few frequencies of their respective graph spectrum without observing any significant performance drop. We believe that this property is specific to the data for community detection, as explained in Sec 3.2. Furthermore, we show (a) that GCN mainly exploits the first eigenvalues of a Laplacian (b) that the high frequencies are less informative and (c) that a GCN doesn’t compute more complex invariants than an MLP fed with spectral features.
>
> We believe those contributions are novel, and that they might impact the community, as datasets like Cora, Citeseer, Pubmed are often used to perform ablation studies, like the oversmoothing phenomenon (Li et al 2018, Rong et al 2019). Oversmoothing makes the exploitation of high-frequencies by a model difficult. However, we show that the informative spectral components for the community detection task are in the low-frequency domain. Thus, for studying this effect, it could be judicious to use different datasets, for which high frequencies are more crucial for a supervised task.
>
> If the reviewer is aware of other works conducting this type of experiments and drawing this type of conclusion, we would be extremely pleased to be linked to those references.
>
> -   General comment: we asked a native speaker to proofread the paper. We firmly apologize for some of the poor writing that we hope to have corrected.
> -   Sec1
> 	-  We agree with the reviewer and we simplified ‘Graph principle’ to ‘principle’.
> 	- As explained above, we clarified our contribution thanks to the clear view of the reviewer.
> 	- ‘Informative’ here stands mainly for greatest change in accuracy, yet we do believe that change in loss and change in accuracy are quite interlaced in this context.
>
> -   Sec3
> 	-  Thanks for pointing out the issue with the notations. We decided to use exactly the notations of (Kipf & Welling 2016), and we hope now any confusion is avoided.
> 	- We added Sec 3.2 which clearly corresponds to a potential framework, as it justifies our low-rank approximation methodology.
>
> -   Sec4
> 	- We corrected the sentence about Pubmed.
> 	- Theoretical justification: we believe Sec 3.2 gives a justification of the well-foundedness of a Graph Laplacian low-rank approximation thanks to a Stochastic Block Model. This generative model is a simple model for understanding community detection tasks in an unsupervised context. In a few words, the informative spectrum band for a community detection task is concentrated in the first frequencies of a Laplacian. We believe the same ideas hold for our work, as the datasets that we used have an underlying community structure.
>
> 	- Fig2: This is an interesting point which is also raised by AnonReviewer1. The drop of performance on Cora for 8-layer GCNs is mainly due to the fact that deeper models are difficult to train, as their training is known to be unstable (Li et al 2018, Rong et al 2019). We could have partially solved this issue by fine-tuning some of our parameters for deeper GCNs, but we believe it would make the comparisons between different models and datasets less clear. Furthermore, note from Fig 6 that spectral perturbations might impact in a heterogeneous manner the GCN: the oscillations we observe are a data-dependent phenomenon which is difficult to avoid.
>
> 	- Method: we believe that now, thanks to your relevant comments, the focus of the paper is clearer: studying low-rank approximations of a graph Laplacian, for community detection tasks. The goal of MLP experiments is to show that a GCN doesn’t capture more complex invariants than an MLP fed with spectral information. We absolutely agree that the topology of the network is encoded in those eigenvectors, as explained in Sec 3.2, and we believe we made a clear point in sec 3.2 about this type of concatenation, which encodes the geometry of this task.
>
> We would be extremely interested to have your view on our modifications and clarifications and we thank you again for your time.

---

### Official Review · AnonReviewer2 · 2020-10-29
**Novelty is limited**

**Rating:** 4
**Confidence:** 5

**Review:**

Brief summary

This paper studies the contribution of low-frequency eigenvalues and high-frequency eigenvalues of graph Laplacian empirically. The authors show that high-frequency eigenvalues are less informative. In their work, they propose an MLP model that can achieve equal or better performance than GCN by using eigenvectors as part of input features.

Pros:

1. MLP fed with node features and spectral coordinates of a given node is competitive with a GCN and more efficient than GCN.
2. The empirical study of high-frequency and low-frequency eigenvalues provides great insights on whether we should build high-pass filters for graph signals.
3. The authors show that high-pass filters tuned properly can lead to some minor performance boosts on GCNs.

Cons:

1. Although it can be done in the pre-processing step, the computation cost of eigenvalue decomposition is heavy. Besides, as the proposed method assumes the corresponding graph topology is fixed, it can not be used in an inductive setting.
2. The idea of feeding node features and spectral coordinates to MLP is not novel. It is a kind of feature engineering.
3. The impact of high-frequency eigenvalues on graph-level tasks is not studied. Graph-level tasks are equally important as node-level tasks for a powerful GCN.
4. The authors said larger eta emphasizes higher frequencies, and when eta=2 or eta=3 it marginally boosts the performance of GCN on CiteSeer and Pubmed. Larger eta not only emphasizes higher frequencies but also lower frequencies because when eta goes to infinity, it leads to an all-pass filter which includes both high frequencies and low frequencies.
Originality

The paper provides new insights about low-pass filters and high-pass filters on graphs. Since the paper essentially proposed to use new features for an mlp model. The novelty of this paper is limited.

Minor comments:

1. About Equation 7 and Equation 8, the illustration is not clear to me. What is the range of eta? going from 0 to infinity? or going from negative infinity to 1? It seems quite ambiguous. In addition, if the range of eta varies from 0 to infinity, I think Equation 8 balances a trade-off between a smoothing low-pass filter and an all-pass filter. It is better to plot the spectrum of Equation 7 to verify you argument.
2. How is Equation 9 derived?

---

> ### Author Response · Authors · 2020-11-13
> **Re: Novelty is limited**
>
>
> Dear Reviewer, we thank you very much for your insightful comments. We have adapted the structure of the paper to address the cons you mentioned and we’d like to answer each point you raised.
>
> Originality: We refocused more precisely our paper on the spectral ablation experiments. Indeed, we believe that we are the first work to conduct a systematic spectral analysis of deep models on community detection tasks and to link it with a low-frequency property of the data. Other works (Li et al 2018, Rong et al 2019) focus typically on the architectures, rather than on the data via ablation studies, contrary to what we do.
> We show (a) that the important spectral components for the supervised community detection task are essentially in the low-frequency domain (b) we conclude that GCNs do not compute more complex invariants than a MLP fed with low-frequencies, in the context of community detection. We have added Sec 3.2 that relates those observations to the well known Stochastic Block Model.
>
> Cons:
> 1.  (computational complexity) We added a paragraph about the cost, that we believe isn’t an issue here. Note that the method is actually quadratic in the size of the graph because one does not need (as shown Fig 5) to perform a full exact SVD, however, as now pointed in the paper, approximate algorithms exist for sparse truncated SVD which are significantly faster (see: [https://arxiv.org/pdf/1607.03463.pdf](https://arxiv.org/pdf/1607.03463.pdf) ). To give a more precise order of magnitude, the calculation of 100 eigenvalues takes respectively 2.5s on Cora, 0.5s on Citeseer, and 14s on Pubmed on a standard laptop.
> (inductive setting) We believe that community detection tasks are still an important setting, and we have emphasized now in our paper (by changing the title, rewriting the contributions, and adding Sec 3.2) that we only focus on this task. We believe this is of high interest as major works (Li et al 2018, Rong et al 2019, Kipf & Welling 2016) often consider community detection tasks and consider Cora, Citeseer, and Pubmed datasets.
> 2.  (feature engineering) We agree with the reviewer that this is simply a form of handcrafted feature. However, we couldn’t find a clear reference that benchmarks an MLP approach combined with spectral features. Furthermore, we re-emphasized that the goal of those experiments is to show that in this context, GCNs do not build a more complex representation than a simple MLP fed with spectral features. It would be extremely kind if the reviewer could point us to a reference that applies this technique to modern deep learning datasets for graphs.
> 3.  (graph-level tasks) We would like to highlight that we clarified in (1) and in the manuscript that the exclusive focus of this paper is the community detection tasks.
> 4.  (eta and all-pass filter) We completely agree with the reviewer. We clarified that different values of $\eta$ might emphasize higher frequencies compared to lower frequencies which are almost not affected. Note that this is very similar to the signal processing case: if $\phi$ is a low pass filter(say a Gaussian filter), then by analogy, we could consider a filter $\phi_\eta(x) = \frac{1}{\eta} \phi(\frac{x}{\eta})$, with corresponding Fourier transform $\hat{\phi_{\eta}}(\omega) =\hat{\phi}(\omega \eta)$ and thus, for small values of $\eta$(e.g., $\eta<1$), a convolution with $\phi_{\eta}$ will maintain more high-frequencies. Given this is a fairly simple argument, we think a plot is not necessary.
>
> Minor comments:
>
> -   Thanks for pointing this out. Before $\eta$ was in the range $[0,+\infty[$. In order to highlight that some values of $\eta$ lead to a degenerate averaging (because its spectrum could be negative), we decided to replace $\eta$ by $\frac{\eta-1}{2}$. This emphasizes the fact that if $\eta<0$, then $\tilde A_\eta$ is a degenerated averaging. As said above, we absolutely agree that a large value of eta impacts both the low-frequencies and high-frequencies, but as Eq (9) shows, the most impacted frequencies are the high-frequencies, because low frequencies are preserved. We emphasized it again in the text.
> -   Equation (9) is derived by considering a Taylor expansion for $\eta\rightarrow \infty$. As requested, we wrote the full proof in the Appendix.
>
> We would be extremely interested to have your view on our modifications and clarifications and we thank you again for your time.

---

### Author Response · Authors · 2020-11-13
**General response**

We thank all the reviewers for their insightful comments that helped to improve the general quality of this manuscript. In particular:

 1. We re-emphasized that we mainly study community detection tasks through ablation studies
 2. We clarified our contributions
 3. We added Sec 3.2 which gives a justification of the success of a low-rank approximation method for a graph Laplacian thanks to the Stochastic Block Model, in the context of community detection

Moreover, we now use the notations of (Kipf & Welling 2016) and we proofread our manuscript by a native speaker to remove the undesirable language mistakes.
Thanks to the OpenReview format, we would be very happy to initiate a discussion with the reviewers if some elements were not addressed satisfactorily, and we believe that a study specific to the standard datasets Cora, Pubmed, and Citeseer can help to understand better the success of several deep learning methods for graphs, and thus that this work has a significant value.
Our major modifications are at the end of Sec 1 and Sec 3.2.

---

### Decision · Program_Chairs · 2021-01-07
**Final Decision**

**Decision:**

Reject

**Comment:**

Though the observation regarding the importance of the low end of the spectrum is interesting in its own right, the paper would be better substantiated by experiments on more datasets and a more thorough characterization of the paper novelty/contrast to state of the art.